# communications
## engineering

# Energy efficient integrated MEMS neural network for simultaneous sensing and computing

Hamed Nikfarjam [1], Mohammad Megdadi [2], Mohammad Okour[3], Siavash Pourkamali[1] & Fadi Alsaleem [3✉]

Biological systems seamlessly combine multiple functions in lightweight and energy-efficient structures. Such capability in synthetic structures would be desirable in numerous engineering applications such as aerospace, robotics and wearable devices. Here we report an integrated silicon-based structure configured to sense, perform different classification algorithms, and produce an action signal within the same physical layer. The algorithms are coded in the mechanical responses of the sensing elements of multiple coupled micro-electromechanical systems (MEMS), simultaneously capturing acceleration measurements to produce an actuated signal. This all-in-one structure operates with zero circuitry and low power consumption. As a demonstration, we designed and fabricated a network of three MEMS neurons to successfully perform both simple signal classification and activity recognition problems (standing and sitting) with only $9.92 \times 10^{-17}$ kWh and $17.79 \times 10^{-19}$ kWh energy consumption per operation, respectively. Our approach will enable emergent technologies, such as wearable devices, to perform complex computations with power from a single battery charge.

[1] Department of Electrical Engineering and Computer Science, University of Texas at Dallas, Richardson, TX 75080, USA. [2] Department of Mechanical and Material Engineering, University of Nebraska—Lincoln, Lincoln, NE 68588, USA. [3] Durham School of Architectural Engineering and Construction, University of Nebraska—Lincoln, Omaha, NE 68182, USA. ✉email: falsaleem2@unl.edu

Biological systems such as human skin, eagle wings, or octopus tentacles inherently integrate sensing, actuation, and controls to produce unique multi-functionality[1]. Inspired by these biological systems, there is a growing interest in developing smart materials (metamaterial) capable of integrating sensing, actuation, and computation into one structure. The advancement of these smart materials will impact a large variety of applications. It may lead to the development of airfoils that change their aerodynamic profile, robotic skins with a realistic sense of touch, or even a robotic tentacle that can autonomously navigate through a patient's body to perform minimally invasive surgery[1]. Mechanical computer is seen as the solution to enable such advancement[2,3]. An example is using a micro-electro-mechanical system (MEMS) for digital computing. Specifically, logic gates such as AND, NAND, OR, or XOR have been realized using a single MEMS device[4]. However, poor cascading compatibility of the MEMS-based logic gates to realize an integrated-like circuit poses a significant challenge[5]. Inspired by biological systems, which compute at the sensor level via sensory neurons and utilize massively parallel, energy-efficient computing in the brain, this paper demonstrates MEMS neural network hardware that can perform simultaneous acceleration sensing and a classification problem at the same sensor physical layer. This computing architecture eliminates the need for the complex sensors interface and a digital computing unit to perform similar computations. Thus, it led to hardware that performs machine learning methods that are no longer shallow and separate between the sensing and the computing layers.

## Results

A MEMS neural network of three neurons is built to perform simultaneous acceleration sensing and classification tasks, as shown in Fig. 1. Each MEMS neuron consists of one suspended proof mass supported by meandering tethers. Each mass has arrays of electrodes extending outwards, forming parallel plate electrostatic actuators with their adjacent electrodes. Depending on their purpose, the electrode arrays are referred to as "softening electrodes" or "coupling electrodes." The force-displacement characteristics of the electrode arrays make the movement of the proof masses highly nonlinear, replicating a neuron's non-linear behavior in a typical recurrent neural network (RNN)[6]. In this analogy, as shown in Fig. 1b and explained in Supplementary Notes 1, softening electrodes, which resemble the bias terms in a typical neural network, induce a negative electrostatic stiffness which softens the structures. Moreover, coupling electrodes, which resemble the neural network weights, electrostatically couple the elements together. The coupling electrodes also provide sophisticated multi-directional interaction mechanisms between the proof masses. For example, as shown in the red arrow in Fig. 1a, the upward movement of mass 3 (M3) reduces the gap between the interacting electrodes between M2 and M3. This, in turn, with respect to M2, will produce a high downward electrostatic force that pushes mass 2 down.

**Activity recognition.** Detecting and understanding activity recognition of standing and sitting (ARSS) is important, as sitting

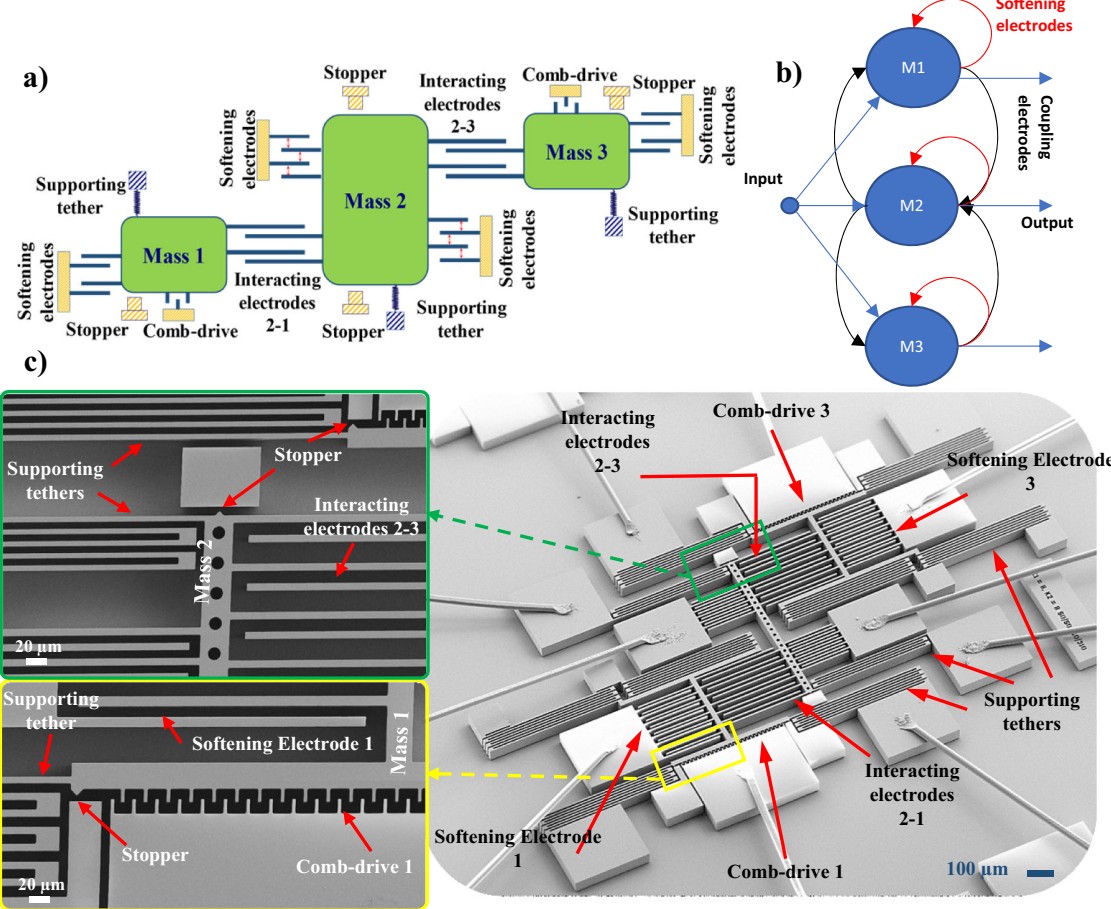

**Fig. 1 The MEMS sensing and neural computing network. a** A schematic showing the three coupled MEMS neurons forming an integrated sensing and computing unit. **b** The analogy of the MEMS computing unit with a neural network. **c** A detailed SEM image of the fabricated MEMS network.

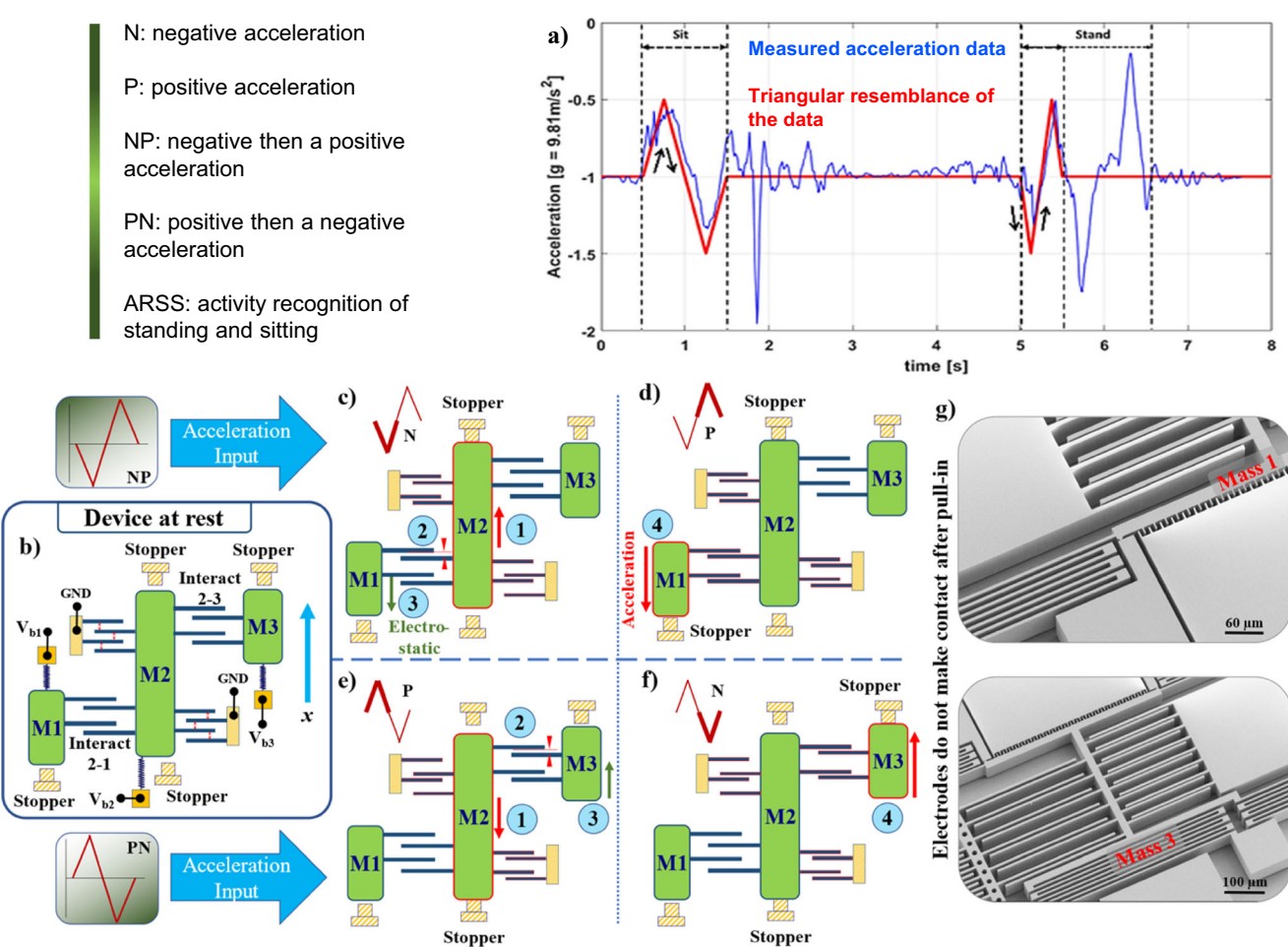

**Fig. 2 A schematic of the 3-MEMS neurons network performing ARSS. a** Shows the measured acceleration data while performing sitting and standing tasks. **b** System at rest with applied bias voltages and no acceleration. From **b**–**d** is the network response to an NP signal, while from **a**, **e**, **f**, is the network response to PN signal. **g** SEM images of the device after pull-in, demonstrating that the electrodes do not make contact due to stoppers.

for long hours is linked to musculoskeletal health problems. As a result, users' sitting behaviors can be improved by using continuous stance monitoring. Wearable devices that run sophisticated algorithms on inertial measurements have been proposed in the literature to provide such monitoring[7]. However, wearable devices generally suffer from the tight power budget requirements to run such techniques continuously in real-life scenarios. Next, we present the use of the MEMS neural network hardware to overcome this challenge.

Observing publicly available data[8] reveals that the acceleration profile, along the vertical direction of a sitting activity, resembles a triangular signal (Fig. 2a) with a positive then a negative (PN) profile. In contrast, the stand activity starts with an opposite pulse (i.e., negative then positive (NP) profile pulse). Thus, one way of performing ARSS is by determining the direction of the acceleration change; if the acceleration is positive then negative, the action is sitting. On the other hand, if the acceleration is negative then positive, the action is standing.

Figure 2b depicts the system at rest with applied bias voltages and no acceleration. Figure 2c illustrates the operation of the MEMS neural hardware as an acceleration classifier for ideal ARSS. In this implementation, M1 should hit its bottom stopper to indicate an NP signal (Fig. 2c, d), or M3 should hit its upper stopper to denote a PN signal (Fig. 2e, f). Each action is considered as a switch closing two different simple circuits representing the different types of the applied signal. To operate the device for ARSS, softening electrodes and comb-drive actuators of elements

1 and 3 have the same bias voltage as their respective masses. Therefore, exerted force from these two components in each element is equal to zero. Softening electrodes of M2 are biased, resulting in having the lowest stiffness in the network. Using this configuration, the classification steps for the NP signal are as follows: (1) as the downward (negative) acceleration part is applied to the fixed components, the suspended parts of the structure will move upward due to inertia. M2, with the lowest stiffness, will dominate the movement and pulls-in toward its upper stopper. (2) This sudden movement will also decrease the gap size in interactive electrodes 2-1 and (3) produce a large electrostatic force pulling M1 down, but not enough to cause it to pull in (Fig. 2c). (4) Once the positive acceleration starts, M1 moves further down, leading to its pull-in toward its lower stopper (Fig. 2d). Due to bistability (memory), M2 will not release when the positive acceleration is applied. Moreover, applying only positive acceleration alone is insufficient for M1 to pull in. The sequence of events will be opposite when detecting the PN acceleration signal performed by M2 and M3 as shown in Fig. 2e, f. Specifically, M2 experiences a downward pull-in due to positive acceleration when a PN signal is applied, reducing the gap in interacting electrodes 2–3. The following negative acceleration results in an upward pull-in for M3. Figure 2g displays SEM images of the device after pull-in, demonstrating that the electrodes do not make contact due to the presence of stoppers.

We used a simulation model for the MEMS network equations to validate the network operation as explained in the "Methods"

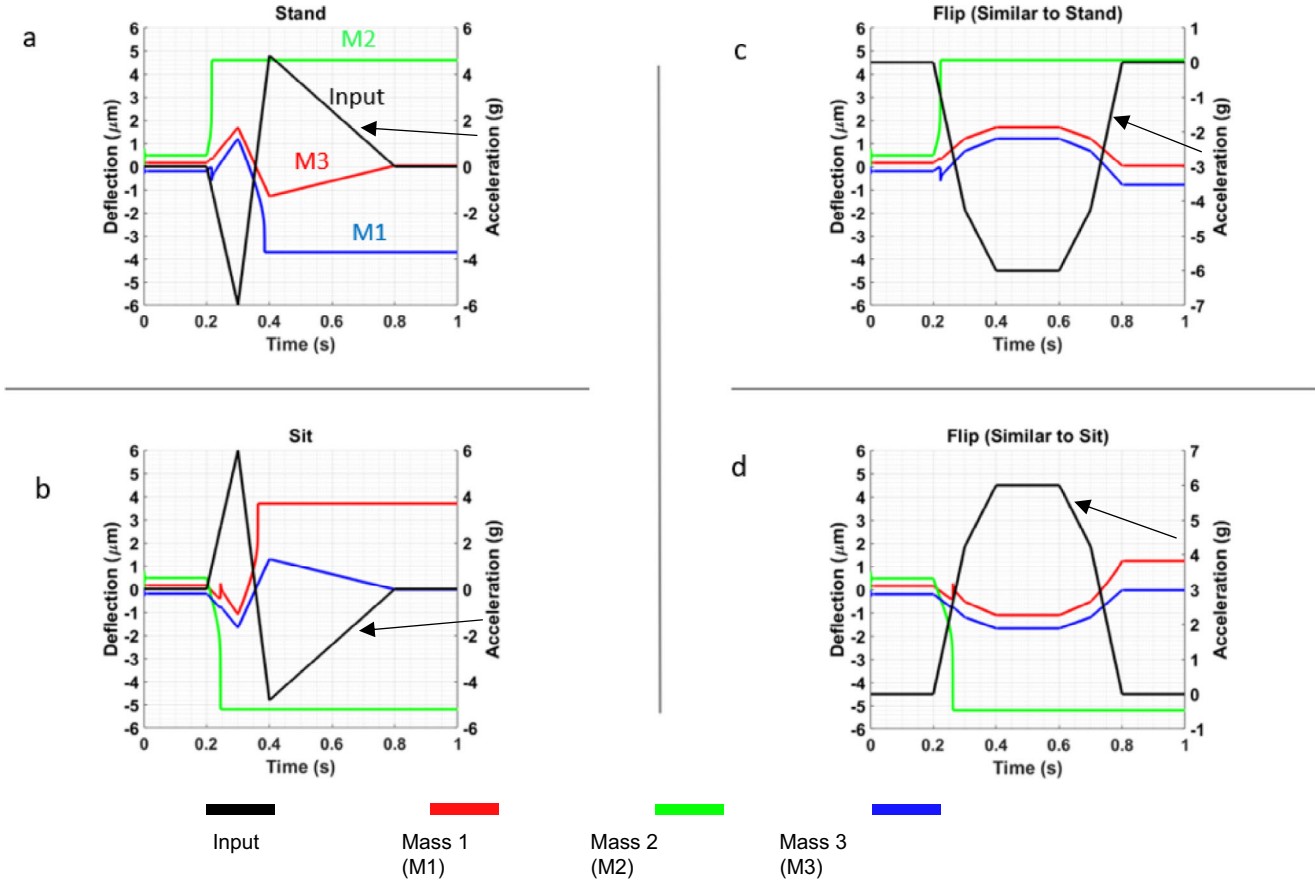

**Fig. 3 Simulation results for the MEMS neural computing unit to perform the sit-stand classification and rejecting false signals.** Simulation for detecting **a** a stand single and **b** a sit signal and rejecting, **c** false stand signal, and **d** false sit signal.

section. Then, we validate some of these simulations using the MEMS hardware presented in Fig. 1. Figure 3a shows the model response where the NP (stand activity) signal is applied. As expected, the Figure shows when the negative part of the acceleration signal is applied, all masses, including the middle mass, green, are moved up. At one point, the middle mass, with the lowest stiffness, will pull up. This sudden movement will increase the electrostatic force to move the lower mass (M1 in blue) down. When the positive part of the input signal acceleration arrives, the electrostatic and acceleration forces together cause the M1 to pull down, indicating the detection of the NP signal. To show the need for consecutive order of negative and then positive acceleration signals for M1 to pull down, we show a false signal of applying only a negative acceleration profile in Fig. 3c. This false signal was selected as it has some characteristics that may confuse with the signal of interest such as the decrease and then increase behavior. But it is different as it does not has the acceleration sign change from negative to positive as is the case for the truth NP signal. Also, the false signal was chosen to have a long period to cover almost the full range of the true NP signal.

Figure 3c shows while at one point after the middle mass is pulled up; the M1 goes down. However, M1 does not move enough to produce a pull-down switching. The simulation results in Fig. 3b show an opposite behavior for the network to cause M1 to pull up when a PN acceleration signal (sitting activity) is applied. Moreover, Fig. 3d shows that the network, similar to Fig. 3c, can reject a false-only positive acceleration signal.

Another simulation study we conduct is by varying the input acceleration amplitude. As an example of this study, Fig. 4 shows

the response of M1 at a peak-to-peak of 5.4 g, 9 g, and 18 g for the standing acceleration profile case. Those values were selected to be below, slightly below, and above the signal amplitude in Fig. 3a. As shown in Fig. 6S in Supplementary Notes 2, in all cases, the middle mass will pull up, as shown in Fig. 3a. The Figure shows that the 5.4 g peak-to-peak signal will not cause the pull-in for M1. Thus, besides the input signal shape, its amplitude is another factor the network can be trained for to perform classification. Adjusting the network sensitivity to the input signal amplitude can be tuned using the bias voltage electrodes.

The experimental testing of a microfabricated MEMS neural unit is presented in Fig. 5. A schematic diagram showing the applied electrical connections is shown in Fig. 5a. Figure 5b shows the device at rest under an optical microscope. Figure 5c, d show the final response of the system when complete NP and PN acceleration signals are applied, respectively. As expected, the Figure shows M2 is pulled upward, and M1 is pulled downward for the NP signal. In this case, the pull-in of M1 indicates the detection of an NP signal. When the PN signal is applied, M2 is pulled downward, and M3 is pulled upward. The pull-in of M3 indicates the detection of a PN signal. To evaluate the ability of the neural computing unit to reject false signals, similar to the simulation results in Fig. 3c, d, two acceleration signals were applied. A positive acceleration signal with a positive slope, then a negative slope. And a negative acceleration signal with a negative slope and then a positive slope, as shown in Fig. 5e, f. These acceleration signals were generated by rotating the device either 90 or −90 degrees from the horizontal position. The right parts of Fig. 5e, f show that under the applications of those false signals, while M2 pulled in upward or downward, none of the other

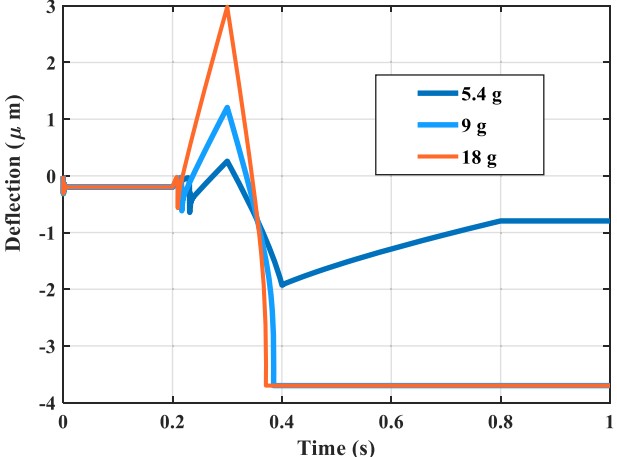

**Fig. 4 Simulation results for M1 of the MEMS neural computing unit to perform the stand classification with different input acceleration signals amplitude.** The Figure shows with an input acceleration of 5.4 g; the network fails to perform the stand classification.

devices pulled in. This indicates that the neural computing unit has correctly rejected those signals.

**Signal classification**. The same hardware was configured differently by changing the bias voltages to perform a different classification problem. This configuration distinguishes between gradually ramping (triangle) and abruptly changing (step/square) input signals[9]. Figure 6 illustrates the operation of the described network as an electrical signal classifier, while Fig. 6a depicts the device at rest with applied bias voltages. Compared to the previous acceleration classification, the signal to be classified is entered as an input voltage to comb-drive actuators of M1 and M3 and the status of M2 determines the signal class; downward pull-in of M2 signals the detection of an abrupt signal, and upward pull-in of M2 signals the detection of a ramp signal. To achieve such an operation, the computing device is configured as follows: (1) Bias voltages applied to all the parallel plate softening electrodes should be set to bring all three elements close to their instability (pull-in) point. (2) M3 is biased in such a way that its potential difference with respect to M2 and, consequently, their interaction force is stronger compared to that of M1 and M2 (M2

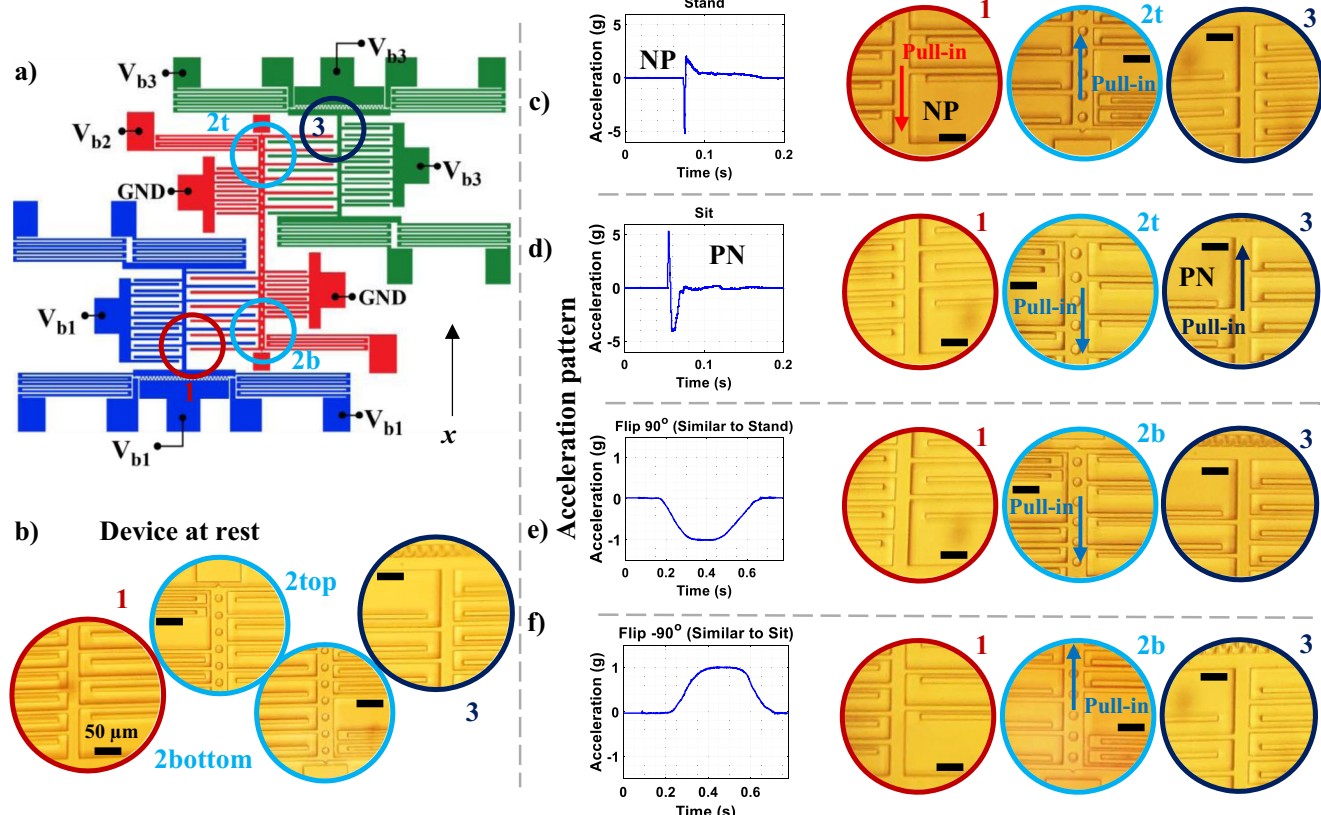

**Fig. 5 Schematic diagram and experimental results of the MEMS network hardware operating in activity recognition mode.** All microscopic images share the same scale bar **a** schematic diagram showing applied bias voltages. $V_{b1} = V_{b2}$ in this configuration. **b** Optical microscope view of separate areas of the device at rest. **c** Simulating the stand-to-sit activity by gently hitting the device in x direction: initial positive acceleration causes suspended mass 2 to move in the negative x direction with respect to the fixed components and a downward pull-in occurs in device 2. The downward movement of mass 2, reduces the upper gap between the 2–3 interacting electrodes. Subsequent negative acceleration results in a positive x displacement of device 3 and a further reduction in the upper gap, causing an upward pull-in for device 3. **d** Simulating the stand-to-sit activity by gently hitting the device in negative x direction: initial negative acceleration causes mass 2 to move in the positive x direction with respect to the fixed components and an upward pull-in occurs in device 2. The upward movement of mass 2, reduces the lower gap between the 2-1 interacting electrodes. Subsequent positive acceleration results in negative x-direction movement of device 1, further reduction in the lower gap and a downward pull-in for device 1. Applying only a negative (by rotating the device 90° from the horizontal position) or only a positive acceleration (by rotating the device −90° from the horizontal position), although causes mass 2 to pull in upward (**e**) or downward (**f**) but is not enough to cause pull in for device 1 or 3.

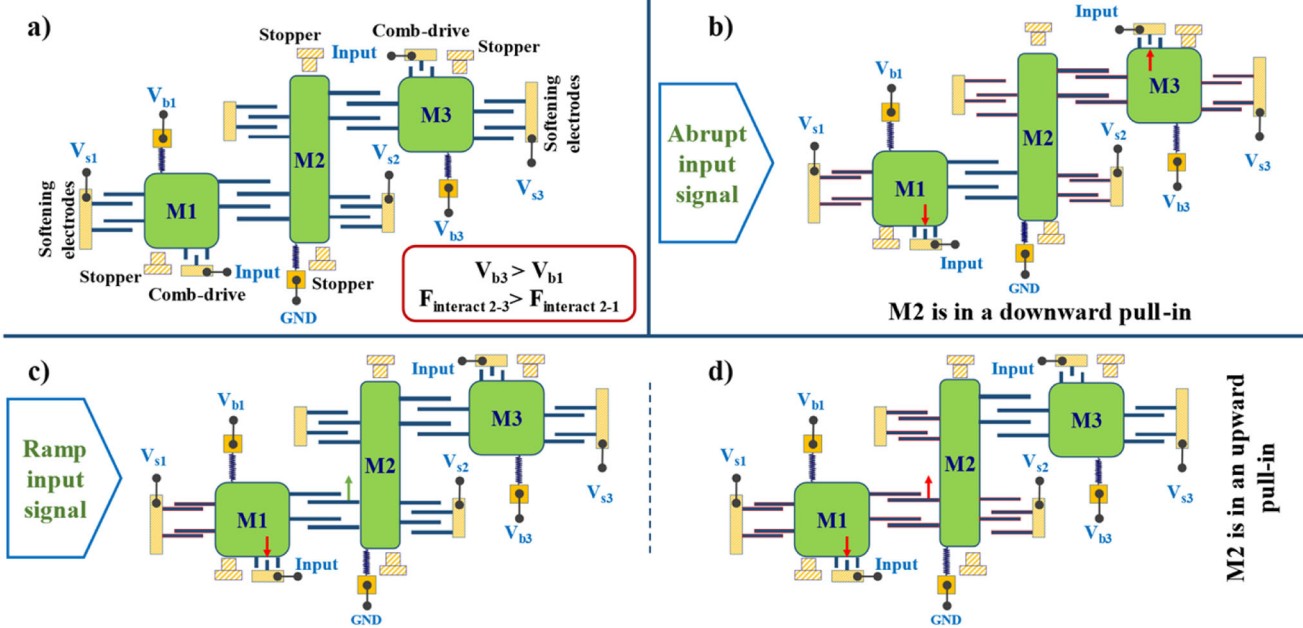

**Fig. 6 A schematic for the operation concept of the MEMS network hardware performing signal classification.** The blue-colored texts indicate the applied bias voltages. **a** Schematic of the device along with applied bias voltages. Since $V_{b3}$ is larger than $V_{b1}$, the interaction force between M2 and M3 is larger than between M2 and M1. **b** An abrupt signal is applied to the inputs and M1 and M3 pull-in simultaneously. A larger interactive force causes M2 to move downward. **c** A ramp signal is applied, and M1 pull-in first as the exerted force from comb-drive 1 is larger. Finally, in panel **d**, the downward pull-in of M1 decreases the gap size in the interacting electrodes 2-1. As a result, M2 is pushed out of stability and an upward pull-in occurs.

is grounded and $V_{b3} > V_{b1}$). (3) However, with similar input applied to the two comb-drive actuators, comb-drive 1 exerts more force on M1 compared to comb-drive 3 on M3 because of the lower voltage difference between M3 and comb-drive 3 ($V_{input}-V_{b1} > V_{input}-V_{b3}$). Under such conditions, if a large enough abrupt voltage is applied to the input comb-drives, both M1 and M3 would pull in almost simultaneously (M1 pulls downward and M3 pulls upward). In this situation, however, the stronger coupling (larger interacting force) between M3 and M2 pulls M2 down towards its lower stopper, indicating the detection of an abrupt increase in the input signal (Fig. 6b). On the other hand, when a gradually increasing signal is applied to the two comb-drives, at one-point M1 pulls in first since exerted force from comb-drive 1 is larger and creates greater gradual displacement compared to comb-drive 3 (Fig. 6c). Downward pull-in of M1 decreases the gap size in the interacting electrodes 2-1, resulting in a larger attraction force that is enough to push M2 out of stability and pull it in the upward direction, hitting its upper stopper. M2 (output neuron) pulling upwards indicates the detection of a ramp signal. As the input voltage keeps increasing (Fig. 6d), eventually, M3 pulls in upward. However, its exerted force on M2 from their interacting electrodes will not be adequate to pull M2 out of the upward pull-in (the system has memory). Figure 7 shows the validation of this operation using the MEMS neural network hardware. Specifically, a schematic diagram showing the applied electrical connections is shown in Fig. 7a. Figure 7b shows the device at rest. Figure 7c shows the final response of the system when an abrupt signal is applied, and finally, Fig. 7d, e show the network response to the ramp signal.

**Discussion**

This paper presented an integrated sensing and computing hardware to perform classification problems. The presented

MEMS computing hardware can be reconfigured to perform completely different classification problems by changing its bias voltages. The classification algorithms are coded in the mechanical responses of the sensing elements of multiple-coupled electrostatic MEMS devices that simultaneously capture the measurement of interest, such as acceleration. As a result, the MEMS computing unit, with zero circuitry need, will perform computing at the sensing physical layer (substrate) and will require very little energy for only the bias voltages (around $9.92 \times 10^{-17}$ kWh for the ARSS and $1.779 \times 10^{-19}$ kWh for the signal classification problem (Supplementary Notes 3) by: (1) eliminating the need to engage energy-hungry circuitry for conditioning and reading sensor outputs, and (2) due to the capacitive electrostatic actuation, the total amount of energy consumed in each classification cycle by the MEMS network is very insignificant. Specifically, the only energy needed to operate the described MEMS network is the energy needed to charge the capacitances associated with the electrostatic actuators. The energy stored in a charged capacitor is given by $Q = \frac{1}{2}CV^2$. When the device is settled in a certain state, not undergoing any change of state, no capacitance charge/discharge occurs. With the leakage current of the silicon capacitors being practically zero, there will be zero power consumption (zero static power consumption). In dynamic situations (time history) when the device is frequently reset and new operations are performed, the energy required for each measurement is still low and can be calculated by the capacitor energy storage equation mentioned above, and the power consumption will be the product of measurement (operation) frequency and energy per operation.

As discussed in Supplementary Notes 1, the MEMS computing capabilities resemble the continuous-time recurrent neural networks (RNNs). RNNs, in comparison to the typical feedforward neural networks (FFNNs) have been shown to achieve high accuracy in classification problems that involve time series input

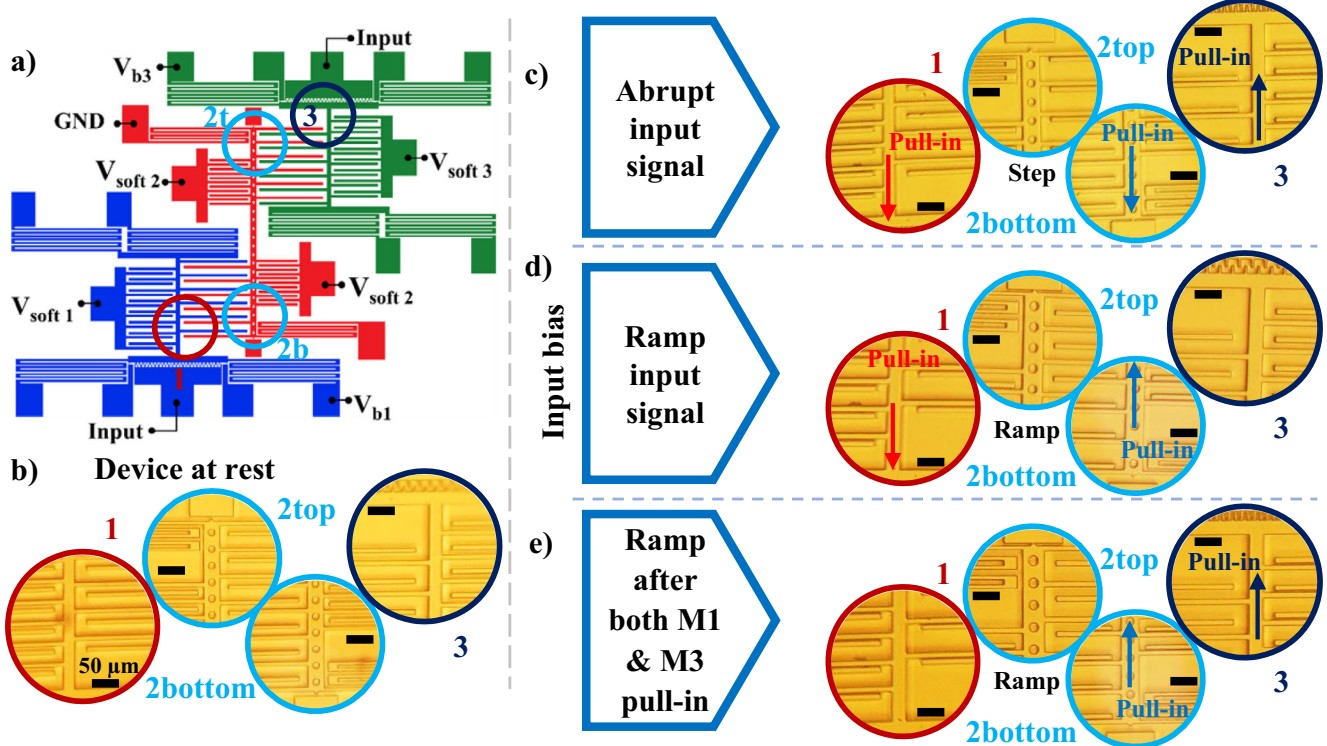

**Fig. 7 Schematic diagram and experimental results of the MEMS network hardware operating a signal classifier.** All microscopic images share the same scale bar. **a** Schematic diagram showing applied bias voltage to different components; **b** Optical microscope view of different areas of the fabricated device at rest; **c** Abruptly increasing the input voltage to 61 volts: both M1 and M3 are pulled-in simultaneously and M2 is pulled towards the element with which it has a stronger coupling (attracted to M3 by a downward pull-in); **d** Ramp input voltage: when input voltage reaches 56 volts M1 and consequently M2 go into pull-in (M1 downward and M2 upward); **e** Ramp signal continued: when the input voltage reaches 61 volts M3 also enters pull-in but does not pull M2 downward.

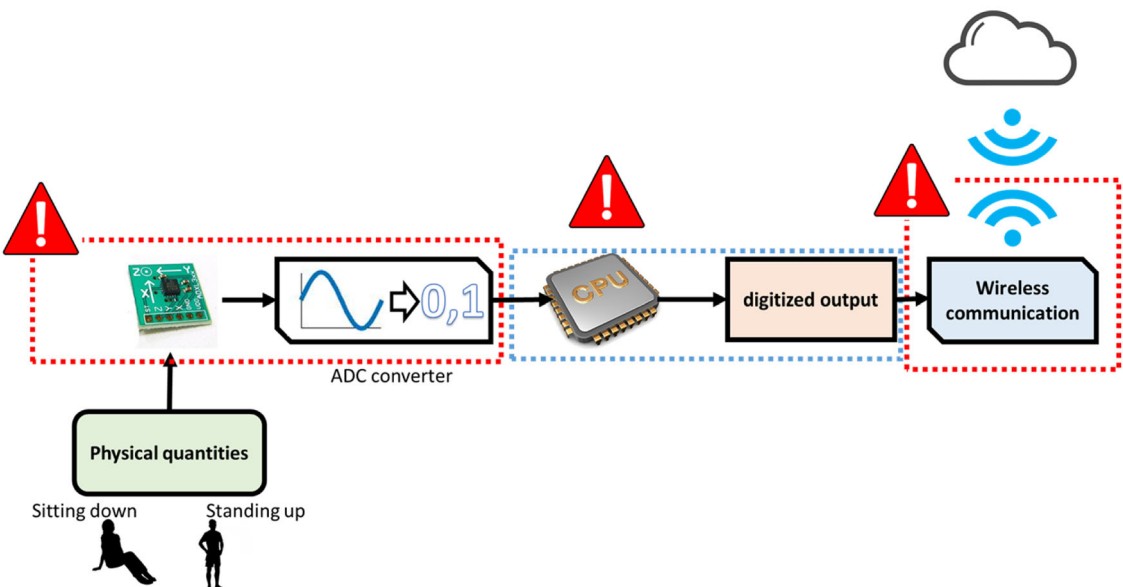

**Fig. 8 A schematic showing the challenge of using a digital computer for wearable device applications to perform classification problems.** Those challenges, marked by the red warning signs in the Figure, are the requirements of: (1) the signal conditioning circuit to convert the physical quantities of acceleration to a voltage signal, (2) converting this signal to a digital signal that is read by a digital computer, and (3) performing the classification locally or transmitting this digitized signal to perform the classification at the cloud. Both operations require a significant amount of the wearable device's limited power.

data. They can process and encode the information contained in the sequential input data[10]. One potential application that can benefit from the presented MEMS computing unit is wearable wrist devices. Figure 8 shows traditional wearable devices with highly sensitive and fast-response MEMS motion sensors, such as accelerometers and gyroscopes, do not have enough processing power to continuously perform RNN or other kinds of machine learning locally due to their stringent power requirement and short battery life[11]. Moreover, the high energy cost of wirelessly transmitting data limits the amount of raw sensor data that can be transmitted to be processed externally[12]. In both cases, due to the need to sample the input acceleration signal, problems such as data delay, loss, or aliasing and lowering the performance of precision systems might occur. Such limitations reduce the accuracy of machine learning models[13].

On the other hand, the presented integrated sensing and computing approach eliminates the need for conditioning the sensor reading and converting it to a digital signal, the power cost of the digital computer to process the data or transmitting the data to be processed by outside computer. To put things in perspective, this cuts most of the power wasted in wearable

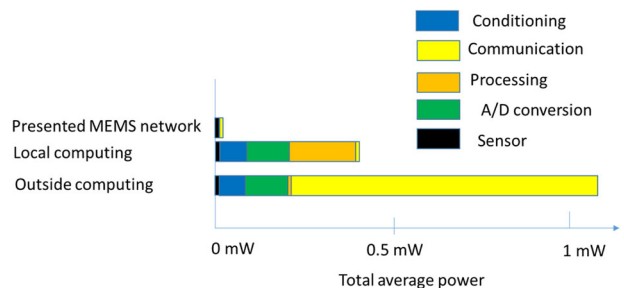

**Fig. 9 Power consumption comparison for sensor measurement between a traditional wearable device and our MEMS computing hardware.** The plot shows the significant amount of power being cut by the MEMS computing network.

devices to perform a classification problem, as shown in Fig. 9. The Figure compares the average power consumed by a wearable device to read a sensor reading, process it locally or transmit it outside[13] to the power to be consumed of the presented MEMS network to perform similar functionalities. This low power consumption will enable the presented MEMS hardware to perform around 0.44 Tera acceleration calcification using a single charge of a common wearable battery (with $4.34 \times 10^{-5}$ kWh capacity). Supplementary Table S1 and Supplementary Table S2 show the estimated energy consumption of the MEMS neural hardware for both the activity recognition (acceleration classification) and the signal classification tasks, respectively.

## Methods

**Fabrication.** SEM view of a fabricated device is shown in Fig. 1c. Elements 1 and 3 are equipped with comb-drive actuators, which, if desired, can exert displacement-independent forces on the vertically moving proof masses. The two elements have identical components. However, they pull-in in the opposite direction if they reach the instability or pull-in point. Depending on the application and implementation, the forces acting on the proof masses can be due to the softening electrodes, interacting electrodes, comb-drive actuators, or acceleration. Therefore, this building block could be utilized as either a signal classifier or an acceleration pattern classifier using only micro-mechanical elements. Elements 1 and 3 are electrostatically coupled to the 2nd element in the middle via two interactive electrodes. Softening electrodes of element 2 have identical gap sizes on each side. All elements have their stoppers to limit the movement of the moving elements and prevent any contact between the electrodes.

A basic single-mask micromachining process was used to fabricate the devices on a highly doped p-type Silicon On Insulator (SOI) substrate with a 50μm thick silicon device layer and 3μm thick buffer oxide layer (BOX). The device layer was first patterned via an optical lithography step, and the structures were carved out of the SOI device layer via deep reactive ion etching (DRIE) to the BOX. The BOX layer was then removed underneath narrower features by a timed hydrofluoric acid etch to suspend the structures. The devices were subsequently immersed in a solution of naphthalene and isopropyl alcohol to mitigate the risk of stiction to the handle layer or between components, utilizing the release technique outlined by Nikfarjam et al.[14].

**Mathematical modeling and simulation.** A mathematical model was built to simulate the described system. The simulation results confirmed the observed

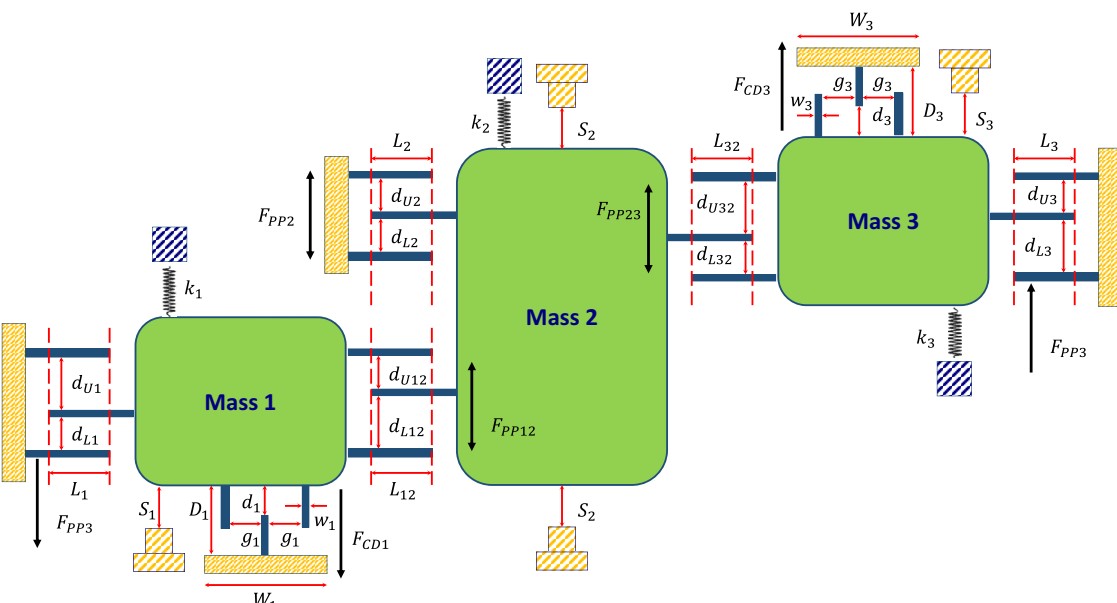

**Fig. 10 Schematic diagram showing the MEMS network model parameters.** $F_{ppi}$ and $F_{ppij}$ are the Parallel-Plates electrostatic force of the ith mass (coupled with fixed plates) and the ith coupled with the jth masses' electrodes. $F_{cdi}$ is the Comb-Drive electrostatic force applied on the ith mass. $d_{Ui}$, $d_{Li}$, $d_{Uij}$, $d_{Lij}$ are the upper gap, lower gap of the Parallel-Plates electrodes either between the ith mass and the fixed plates or the ith and jth masses. $L_i$ and $L_{ij}$ are the overlapping lengths of the Parallel-Plates between the ith and the ground plates or the ith and jth plates. $w_i$, $g_i$, $d_i$, and $D_i$ are the comb-drive finger's width, lateral gap, vertical gap, and vertical gap of the ith mass, respectively.

experimental results. The mathematical model of the device is described in (1):

$$M\ddot{\mathbf{X}} + C\dot{\mathbf{X}} + K\mathbf{X} = \mathbf{F_{CD}} + \mathbf{F_{PP}} - M\ddot{\mathbf{y}} \qquad (1)$$

where the matrices are governed by Eqs. (2) and (3):

$$\mathbf{X} = \begin{bmatrix} x_1 \\ x_2 \\ x_3 \end{bmatrix}, M = \begin{bmatrix} m_1 & 0 & 0 \\ 0 & m_2 & 0 \\ 0 & 0 & m_3 \end{bmatrix}, C = \begin{bmatrix} c_1 & 0 & 0 \\ 0 & c_2 & 0 \\ 0 & 0 & c_3 \end{bmatrix}, K = \begin{bmatrix} k_1 & 0 & 0 \\ 0 & k_2 & 0 \\ 0 & 0 & k_3 \end{bmatrix}$$

$$(2)$$

$$\mathbf{F_{CD}} = \begin{bmatrix} F_{CD1} \\ 0 \\ -F_{CD3} \end{bmatrix}, \mathbf{F_{PP}} = \begin{bmatrix} F_{PP1} + F_{PP12} \\ F_{PP2} - F_{PP12} - F_{PP32} \\ F_{PP3} + F_{PP32} \end{bmatrix}, \qquad (3)$$

where $\mathbf{y}$ is the base absolute displacement and $\mathbf{X}$ is the relative displacement matrix of the masses. $M$, $C$, and $K$ are the mass, damping constant, and stiffness matrices, respectively. $\mathbf{F_{CD}}$ and $\mathbf{F_{PP}}$ are the comb-drive and parallel-plates electrostatic force matrices, respectively, that are governed by:

$$\mathbf{F_{CDi}} = N_{CDi}\varepsilon t V_{CDi}^2 \left( \frac{1}{g_i} + \frac{2w_i}{(d_i - x_i(t))^2} + \frac{W_i - 2N_{CDi}w_i}{N_{CDi}(D_i - x_i(t))^2} \right)$$

$$\mathbf{F_{PPij}} = \frac{1}{2} N_{PPij}\varepsilon t L_{ij} V_{PPij}^2 \left( \frac{1}{\left(d_{Uij} - (x_i(t) - x_j(t))\right)^2} - \frac{1}{\left(d_{Lij} + (x_i(t) - x_j(t))\right)^2} \right)$$

$$(4)$$

where, as described in Fig. 10, $N_{CDi}$ is the number of comb-drive fingers coupled with mass $i$. $V_{CDi}$ is the voltage applied between mass $i$ and the associated comb-drive. $g_i$ is the lateral gap between the coupled fingers, $d_i$ is the vertical gap between them, and $D_i$ is the vertical gap between the comb drive itself and mass $i$. $W_i$ is the width of the comb drive while $w_i$ is the finger's width. $N_{PPij}$ is the number of coupling parallel-plate electrodes between masses $i$ and $j$. And $V_{PPij}$ is the voltage applied between them. $L_{ij}$ is the overlapping length of the electrodes. $d_{Uij}$ and $d_{Lij}$ are the upper and lower gaps between the sandwiched parallel-plates electrodes, respectively. $t$ is the silicon structure's thickness and $\varepsilon$ is the absolute permittivity of air. $\ddot{y}$ is the acceleration input to the MEMS network. As the MEMS proof mass will move when the acceleration signal is applied, the network will directly respond to acceleration without the need for a signal conditioning circuit as is the case for typical neural networks implanted in digital computers. Finally, the output of this MEMS network is the switching status between mass 3 and its stopper and mass 1 with its stopper.

**Experiment procedure**. The fabricated device was mounted on a PCB and wire-bonded. The applied biases for the acceleration classification can be seen in Fig. 5a. All masses are electrically connected through their conductive supporting springs, and the softening electrodes of element 3 are grounded. As previously stated, for this particular application, the softening electrodes and comb-drive actuators do not exert any force on M1 and M3. M1, M3, their softening electrodes, and comb-drive actuators were biased with 8.5 V (Vb1 = Vb2 = 8.5 V). The pull-in voltage of M2, using the parallel plate softening electrodes, was measured to be 27.6 volts. To generate negative electrical stiffness and increase M2's sensitivity to lower accelerations, Vb2 was set to 27.5 V, which is slightly lower than the pull-in voltage of that element. Additionally, an accelerometer was attached to the PCB to measure the real-time acceleration applied to the device.

In order to achieve the desired signal classification, the device requires a specific application of biases, as depicted in Fig. 7a. Specifically, Vb1 has been set to 26 V, while Vsoft-1 is set to 9 V. Additionally, M2 is connected to the ground, and 28 V is applied to its softening electrodes (Vsoft-2). Vb3 and Vsoft-3 have been set to 36 V and 12 V, respectively. The input voltage controlling the comb-drive actuators was abruptly increased from 0 to 70 volts (step signal) to test the device's ability to detect abruptly increasing signals. For the ramping experiment, the input signal was gradually increased from 24 V to 70 volts.

## Data availability

The authors declare that the data supporting the findings of this study are available within the paper and its supplementary information files.

## Code availability

The MATLAB Code used in this manuscript is available at GitHub https://github.com/Megdady/3MEMS-Elctrostatically-Coupled-Model.git.

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

## Acknowledgements

This work is funded by the National Science Foundation, grant number (#1935598, #1935641) (F.A. and S.P.).

## Author contributions

Conceptualization: F.A. and S.P. Methodology: F.A. and S.P. Experimental design and validation: H.N. Simulation: M.M. and M.O. Funding acquisition: F.A. and S.P. Supervision: F.A. and S.P. Writing – original draft: H.N. and F.A. Writing – review and editing: S.P.

## Competing interests

The authors declare no competing interests.
