## [Peer Review File · Communications Engineering]

Reviewers' comments:

Reviewer #1 (Remarks to the Author):

In this paper, an integrated silicon-based material that sense, computes and actuates were proposed and its performances were evaluated. In addition, the proposed all-in-one smart material consumes near zero power and runs with zero circuitry.

Followings are comments for this paper

1. The subject is interesting. However, the presentation of results to assert the superiority of the proposed system is insufficient. Most of the figures in the main text are for explaining the concept or explaining the operation of the MEMS system. Figure S6 of the supplementary material presents the results, and it would be better to include these figures in the main text.
2. The figure number S26 in the supplementary material appears to be a typo in S6.
3. It is necessary that d and f in Figure 2 also include red arrows explaining the operation in the same way as b and c.
4. It is necessary to indicate the components of the x and y vectors of Equation (1) in Figure 6.
5. Since "near zero power and circuit" is included in the title, it is necessary to include the result of this in the main text.

Reviewer #2 (Remarks to the Author):

1. In the title, the authors used "Smart Material" which has a broad range. This needs to be changed to specific smart material treated in this work (silicone-based material).
2. The key technical contribution of this work is "Near Zero Power". However, the authors did not present any power (or voltage) applied to many different electrodes. The time history applied to the electrodes is required to be presented.
3. Both sensing and actuating characteristics should be presented in various conditions. For example, output sensing signal to different input signal and actuating force to achieve different motion. Then, the potential readers can understand the physical concept of the proposed work.
4. The author asserted in Abstract that the proposed smart material can be applicable to soft robotics and wearable devices to perform complex computations powered by permanent batteries. This assertion needs to be validated via experiment or simulation at least. Otherwise, the authors need to explain specific applications those are matched with the proposed smart material in terms of controllable domain, response time, property change and so forth.

Reviewer #3 (Remarks to the Author):

This paper discuss interesting problem describing the materials with computing possibility.

The following question could improve the work:

- 1) Could you discuss whether the material is applicable in soft robotics or soft devices? From my point of view, most sensors require some external deformation or stress to work.
- 2) In section "Mathematical Modelling and Simulation" would it be possible to add information about the connection between model (1) and neural network. For instance, what is input signal and what is output signal.

3) Fig. S7 (in supplementary) should be formatted in publication standards

4) Fig. S26. does not have names of signals.

5) Would it be possible to add some results of experimental tests? For instance, Fig. S26 shows only simulation results.

Reviewers' comments:

Reviewer #1 (Remarks to the Author):

In this paper, an integrated silicon-based material that sense, computes and actuates were proposed and its performances were evaluated. In addition, the proposed all-in-one smart material consumes near zero power and runs with zero circuitry.

Followings are comments for this paper

1. The subject is interesting. However, the presentation of results to assert the superiority of the proposed system is insufficient. Most of the figures in the main text are for explaining the concept or explaining the operation of the MEMS system. Figure S6 of the supplementary material presents the results, and it would be better to include these figures in the main text.

We agree with the reviewer on this point. We have restructured the paper to better present the results, this includes bringing Fig.6 from the supplementary material (now Fig.4) and adding more simulation figures to the main paper (Fig.5) and other simulation figures to the supplementary material.

2. The figure number S26 in the supplementary material appears to be a typo in S6.

Thank you for catching this typo, Figure 6 is moved to the main paper and that typo has been fixed

3. It is necessary that d and f in Figure 2 also include red arrows explaining the operation in the same way as b and c.

We added a similar schematic explanation to d and f in Figure 2.

4. It is necessary to indicate the components of the x and y vectors of Equation (1) in Figure 6.

All the forces in equation 1 were added to Figure 6.

5. Since "near zero power and circuit" is included in the title, it is necessary to include the result of this in the main text.

We added new discussions and new figures to compare the power consumed by this network to the typical power consumption of a digital computer to handle such computing. The new figures are Fig.9 and Fig.10. Specifically figure 10 shows the near-zero power consumption of this network.

Reviewer #2 (Remarks to the Author):

1. In the title, the authors used "Smart Material" which has a broad range. This needs to be changed to specific smart material treated in this work (silicone-based material).

The title is adjusted based on the reviewer's feedback

2. The key technical contribution of this work is "Near Zero Power". However, the authors did not present any power (or voltage) applied to many different electrodes. The time history applied to the electrodes is required to be presented.

The actual voltages applied to each electrode for each signal classification problem we listed in the last figure in the supplementary material. Though the reviewer is touching on a great point about the power and time history response that is worth more explanation. First of all, the term power consumption is revised to energy consumption per classification. The only energy needed to operate the described devices is the energy needed to charge the capacitances associated with the electrostatic actuators. The energy stored in a charged capacitor is given by $Q = \frac{1}{2} CV^2$. The capacitance in the last figure of the supplementary material was calculated based on the geometry of the design. When the device is settled in a certain state not undergoing any change of state, no capacitance charge/discharge occurs and with the leakage current of the silicon capacitors being practically zero, there will be zero power consumption (zero static power consumption). In dynamic situations (time history) when the device is frequently reset and new operations are performed, the energy required for each measurement can be calculated by the capacitor energy storage equation mentioned above, and the power consumption will be the product of measurement (operation) frequency and energy per operation.

We have addressed this point by expanding the discussion section and adding more figures (Fig.9 and Fig10).

3. Both sensing and actuating characteristics should be presented in various conditions. For example, output sensing signal to different input signal and actuating force to achieve different motion. Then, the potential readers can understand the physical concept of the proposed work.

Per the first reviewer's comments, some of those cases were presented in the supplementary material and should be in the main paper. We have moved these simulations and explained them more in the manuscript. We also added new simulation results, Fig.5, per the reviewer's suggestion. This new figure explores the effect of different excitation signals that may affect the classification.

4. The author asserted in Abstract that the proposed smart material can be applicable to soft robotics and wearable devices to perform complex computations powered by permanent batteries. This assertion needs to be validated via experiment or simulation at least. Otherwise, the authors need to explain specific applications those are matched with the proposed smart material in terms of

controllable domain, response time, property change, and so forth.

We dropped the smart robotics from the listed potential application, but we kept the wearable device. In the paper, we added a detailed discussion about the current limitation of using advanced computing in wearable devices and how this proposed material can overcome some of these limitations. Then we compared the power consumption saving by using the presented MEMS network. This newly added lengthy discussion in pages 9 and 10 is supported by adding two new figures (Fig.9 and Fig.10). Concerning the permanent batteries claim and considering the most common wearable batteries such as the silver-oxide type (<https://www.batteryequivalents.com/wrist-watch-battery-replacement-chart.html>), those produce around 43.34 10⁻⁵ kWh capacity (https://media.digikey.com/pdf/Data%20Sheets/Seiko%20Instruments%20PDFs/SR626SW_DS.pdf). Considering the low energy consumption of the presented MEMS hardware, this implies that around 0.44 Tera classification operations can be performed before the battery dies out. These calculations were added in a new paragraph at the beginning of page 11. added to the

Reviewer #3 (Remarks to the Author):

This paper discuss interesting problem describing the materials with computing possibility.

The following question could improve the work:

1) Could you discuss whether the material is applicable in soft robotics or soft devices? From my point of view, most sensors require some external deformation or stress to work.

Per the second review comment and based on your valid concern we feel using this material for soft robotics, while possible, needs a more detailed explanation beyond the scope of this paper. So we focus our discussion on wearable device applications. We expanded the discussion section to provide more detailed information that compares the presented MEMS network to perform classification to the state of art digital computer in a wearable device. We also added a new figure (Fig.1) to show how this new sensing and computing architecture s a complete departure from the current sensing and computing approaches.

2) In section "Mathematical Modelling and Simulation" would it be possible to add information about the connection between model (1) and the neural network. For instance, what is input signal and what is output signal.

The information about the model and how it compared to atypical neural network implementation by a digital computer was added just before Figure 11 on page 10.

3) Fig. S7 (in supplementary) should be formatted in publication standards

The figure was formed according to the reviewer's feedback

4) Fig. S6. does not have names of signals.

Figure S6 has moved to the main manuscript per the other reviewer's suggestions and more details were added to the figure.

5) Would it be possible to add some results of experimental tests? For instance, Fig. S6 shows only simulation results.

Again per other reviewers' feedback, Fig.S6 should be in the main manuscript. This figure, now as Fig.4,

is in the main manuscript. This figure is the simulation results that compare with the experimental data presented in Fig.6 in the main manuscript. However, as it is hard to monitor the deflection of the three masses of the MEMS network in the experiment, the experimental data in Fig.6 shows the final status of each mass, which matches those in the simulation results in Fig.4. For classification purposes, what matters at the end is those final statuses of the three masses. For example, Fig.6 confirmed the simulation in Fig.4, by showing that when an NP acceleration signal is applied, Mass 1 pulls down, and when PN signal is applied, Mass 3 pulls up.

REVIEWERS' COMMENTS:

Reviewer #1 (Remarks to the Author):

In the revised paper, most of the matters mentioned by the reviewers were well reflected and revised, and the quality of the paper was improved. In particular, the resulting data were well complemented. However, additional supplementation is needed for the following points.

1. Expression methods or fonts such as a), b), c), d) inside the figures are all different, but they must be modified in the same form.
2. The titles of Figures 3 and 7 should be written in the same form as the titles of other figures, with explanations for each a), b), c), d), etc.

Reviewer #2 (Remarks to the Author):

This paper has been well revised based on my review comments. Thus, i recommend it for publication in its current form.

Reviewer #3 (Remarks to the Author):

The revision has significantly updated the work.

Reviewer #1 (Remarks to the Author):

In the revised paper, most of the matters mentioned by the reviewers were well reflected and revised, and the quality of the paper was improved. In particular, the resulting data were well complemented. However, additional supplementation is needed for the following points.

1. Expression methods or fonts such as a), b), c), d) inside the figures are all different, but they must be modified in the same form.

We have made all of those to be the same form

2. The titles of Figures 3 and 7 should be written in the same form as the titles of other figures, with explanations for each a), b), c), d), etc.

We have changed the titles of these figures to match the rest

Reviewer #2 (Remarks to the Author):

This paper has been well revised based on my review comments.

Thus, i recommend it for publication in its current form.

Thank you!

Reviewer #3 (Remarks to the Author):

The revision has significantly updated the work.

Thank you!